# Effects of *Bacillus coagulans* and *Lactobacillus plantarum* on the Fermentation Characteristics, Microbial Community, and Functional Shifts during Alfalfa Silage Fermentation

**DOI:** 10.3390/ani13050932

**Published:** 2023-03-04

**Authors:** Yan Wang, Wencan Ke, Qiang Lu, Guijie Zhang

**Affiliations:** Department of Animal Science, Ningxia University, Yinchuan 750021, China

**Keywords:** alfalfa, *Bacillus coagulans*, fermentation composition, microbial diversity

## Abstract

**Simple Summary:**

Dry matter loss (DM loss) occurs during silage production, with the potential to reach up to 4–20% due to the aerobic respiration. Lower water-soluble carbohydrates (WSC) and a higher buffering capacity can contribute to this; however, the presence of *Bacillus* can help create an anaerobic environment, which is more conducive to the growth of anaerobic bacteria. The objective of this study was to evaluate the potential of *Bacillus coagulans* (BC) as an inoculant for silage fermentation. The results showed that adding BC increased the fermentation quality of alfalfa silage, especially when applied together with *Lactobacillus plantarum* (LP). This was evidenced by a reduction in the neutral detergent fiber (NDF) and acid detergent fiber (ADF) contents, as well as an increase in *Lactobacillus* abundance and a decrease in *Enterococcus* abundance after 60 d of fermentation. Additionally, the application of LP, BC, and their combination stimulated cofactor and vitamin metabolism abundance, whilst suppressing drug resistance: antimicrobial pathway abundance. Thus, BC could be considered a viable bioresource for improving fermentation quality.

**Abstract:**

This study aimed to investigate the potential of *Bacillus coagulans* (BC) as an inoculant in alfalfa silage fermentation. Fresh alfalfa was harvested at a dry matter (DM) content of 329.60 g/kg fresh weight (FW), and inoculated without (CON) or with BC (1 × 10^6^ CFU/g FW), *Lactobacillus plantarum* (LP, 1 × 10^6^ CFU/g FW), and their combinations (LP+BC, 1 × 10^6^ CFU/g FW, respectively). Samples were taken at 3, 7, 14, 30, and 60 d, with three replicates for each. The prolonged ensiling period resulted in a decrease in pH values and an increase in lactic acid (LA) concentrations in alfalfa silages. After 60 d of fermentation, the application of BC and LP decreased the pH values and increased LA concentrations in treated silages, especially when their combination was applied. Application of BC preserved more water-soluble carbohydrates (WSC), and further application of BC increased WSC in LP+BC-treated silage compared to LP-treated silage. There was no significant difference in the crude protein (CP) content between the CON and treated silages, however, the BC and LP treatments reduced the ammonia nitrogen (NH_3_-N) concentration, especially when their combination was applied. Additionally, the BC and LP-treated silages had lower neutral detergent fiber (NDF) and acid detergent fiber (ADF) when compared to the CON silage (*p* < 0.001). Inoculants also increased *Lactobacillus* abundance and decreased *Enterococcus* abundance after 60 d of fermentation. Spearman’s rank correlation analysis revealed a positive correlation between LA concentration and *Lactobacillus* abundance. It was noteworthy that LP, BC, and their combination increased the relative abundances of carbohydrate metabolism, energy metabolism, cofactors, and vitamin metabolism, decreasing the relative abundances of amino acid metabolism and drug resistance: antimicrobial. Therefore, the inclusion of BC increased the fermentation quality of alfalfa silage, with the optimal combination being LP+BC. According to the findings, BC could be considered a viable bioresource for improving fermentation quality.

## 1. Introduction

During each stage of silage production, there is a certain amount of dry matter loss (DM loss), which reduces the silage quality. Notably, the DM loss of aerobic respiration could be up to 4–20% [1], and even 2% higher in alfalfa silage than in grass silage due to the lower water-soluble carbohydrates (WSC) and higher buffering capacity [2]. Although a number of lactic acid bacteria (LAB) have been developed to reduce DM loss in silage fermentation, traditional LAB inoculants primarily target the anaerobic fermentation phase and have little effect on DM loss in the aerobic stage [3,4]. The development of silage inoculants is still ongoing, shifting from homofermentative LABs such as *Lactobacillus plantarum* and heterofermentative *Lactobacillus buchneri* to LAB strains with prebiotic functions [5]. Therefore, the strains that could consume oxygen, shorten the aerobic stage, and reduce DM loss also have the potential to be silage inoculants.

Previous studies have demonstrated that *Bacillus* could consume free oxygen and generate an anaerobic environment, which favors the growth of anaerobic beneficial intestinal bacteria, such as *Bifidobacteria* and *Lactobacilli*, and improves animal performance when administered orally [6,7]. Currently, *Bacillus* have been used as a silage additive to enhance the fermentation quality and aerobic stability of alfalfa and corn silage [8,9,10]. Of the *Bacillus* species, *Bacillus coagulans* (BC) has become a focus of research due to its lactic acid-producing capability, degradation of lignin model compounds [11], and strong resistance to acid, heat, and pressure [12,13]. Additionally, BC has also been used as a feed supplement for monogastric animals, with satisfactory growth performances observed in pigs, fishes, and chickens [14,15,16]. However, few studies have assessed its effects on silage fermentation characteristics. We hypothesized that BC could shorten the aerobic phase of silage by consuming free oxygen during the initial stage, finally improving silage quality.

Silage fermentation is a process involving the interactions of microbes, and a better understanding could help to regulate silage fermentation. With the progress of science and technology, microbiomics technology has been used to provide specific microbial information in the silage ecosystem [17,18]. It has only been in recent years that this new method has been widely applied to the study of silage microbial communities [19]. To the best of our knowledge, few studies have confirmed the potential of microbes to be inoculants by using microbiomics technology. Therefore, this study investigated the effects of BC on silage fermentation characteristics and microbial community, confirming the potential of BC as an inoculant in alfalfa silage fermentation.

## 2. Materials and Methods

### 2.1. Materials and Silage Preparation

Alfalfa (cultivar “WL 440”) was harvested at the early bloom stage from three randomly selected plots of Ningxia University located in Yinchuan city, Ningxia province, China. Fresh alfalfa was wilted naturally to the dry matter (DM) content of 329.60 g/kg (within 8 h) and chopped into 1–2 cm pieces immediately using a hay cutter. The chopped forages from each plot were divided into 21 equal piles (each weighing approximately 500 g). One pile was immediately frozen at −20 °C for further analysis. The remaining 60 piles (5 ensiling durations × 4 treatments × 3 replications) were randomly assigned to one of the following treatment groups: (1) distilled water control (CON), (2) *Lactobacillus Plantarum* (LP, provided by Ningxia University, Yinchuan, China), (3) *Bacillus coagulans* (BC, provided by China center of industrial culture collection, identification number: CICC21735, Beijing, China), and (4) *L. Plantarum* plus *B. Coagulans* (LP+BC). The inoculants were applied at a rate of 1 × 10^6^ CFU/g FW, and an equal volume of distilled water was used for the CON. All silage were vacuum packaged in polyethylene plastic bags using a vacuum packaging machine (Zhucheng Yizhong machinery Ltd., Shandong, China). The alfalfa silage bags were stored at room temperature (25 ± 2 °C) and sampled at 3, 7, 14, 30, and 60 d.

### 2.2. Analysis of Chemical Composition, Fermentation Characteristics and Microbial Composition

The DM content was determined by drying the fresh and ensiled forages at 65 °C in an oven for 72 h and then ground with a mill (1 mm screen). Crude protein (CP), ether extract (EE), neutral detergent fiber (NDF), and acid detergent fiber (ADF) were analyzed according to the methods of the Association of Official Analytical Chemists (AOAC) [20], while ammonia nitrogen (NH_3_-N) and water-soluble carbohydrate (WSC) were measured according to Cai [21].

A 20 g of sample was placed in a juice extractor (BA-828, Mannengda Plasthetics Co., Ltd., Guangdong, China) and squeezed for 30 s with 180 mL of distilled water. The filtrate was then filtered through four layers of cheesecloth, and the pH was measured using a glass electrochemical pH meter. Lactic acid (LA), acetic acid (AA), propionic acid (PA), and butyric acid (BA) were measured by high-performance liquid chromatography (KC-811, column, Shodex, Shimadzu, Japan; oven temperature, 50 °C; flow rate, 1 mL/min; SPD, 210 nm) [22].

Samples (5 g) of fresh forage and silage were blended with 45 mL of sterile water and diluted (10^−1^ to 10^−5^) after being shaken for 0.5 h at room temperature. To analyze the microbial composition, LAB was cultured on MRS medium in an anaerobic box (ANX-1; Hirosawa Ltd., Tokyo, Japan) at 37 °C for 48 h, whereas yeasts and molds were cultured on potato Dsandy AGAR at 28 °C for 72 h [21]. The number of viable microorganisms per gram of fresh matter (FM) was calculated in log_10_ CFU.

### 2.3. Microbial Diversity Analysis

For molecular analysis of microbial communities, genomic DNA was extracted using a FastDNA^®^ SPIN Kit (Tiangen, DP302-02, Tiangen, China). Th e DNA quality was verified by an ultramicro spectrophotometer (Thermo, NanoDrop 2000, Waltham, MA, USA). The DNA used for high-throughput sequencing was amplified using primers 338F (5′-ACTCCTACGGGAGGCAGCAG-3′) and 806R (5′-GGACTACHVGGGTWTCTAAT-3′), targeting the V3-V4 regions of 16S rRNA. The PCR was conducted in a 20 μL mixture containing 1.6 μL primer mix (5 μM), 10 ng template DNA, 4 μL 5 × FastPfu Buffer, 2 μL dNTPs (2.5 mM each), 0.4 μL FastPfu Polymerase and 0.2 μL BSA, with dd H_2_O added to reach 20 μL. The PCR products were evaluated with 1% agarose gel and purified with AxyPrep DNA Gel Extraction Kit (Axygen Biosciences, Union City, CA, USA). Amplified fragments were sequenced on an Illumina MiSeq PE300 platform (Majorbio Bio-pharm Technology Co., Ltd., Shanghai, China).

All raw reads were cleaned by discarding those <50 bp and those not matching standard barcodes. The OTUs were clustered, and chimeras were removed based on 97% similarity. Each sequence was annotated for species classification using the RDP classifier, and compared to the Silva database, setting a comparison threshold of 70%, and counting each sample at the taxonomic level of community species composition. Principal coordinates analysis (PCoA) was drawn using the Vegan and Python packages. The linear discriminant analysis effect size (LEfSe) was used to calculate the communities or species with significant differences among the four groups. To investigate the relationship between silage quality and the bacterial community, Spearman’s rank correlation coefficients were generated using R language (version 3.0.2). Using the relative abundances of microbes at the species level, the top 50 independent variables were calculated, and the dependent variables were calculated based on pH, NH_3_-N, LA, and AA. The bacterial function was predicted using the Kyoto Encyclopedia of Genes and Genomes (KEGG) database Phylogenetic Survey of Communities in Unobserved States Reconstruction (PICRUSt2, v2.3.0_b, https://github.com/picrust/picrust2, accessed on 21 August 2022), which predicts the functional abundance of a sample based on the abundance of tagged gene sequences in the samples [23].

### 2.4. Statistical Analysis

Data was analyzed using One-Way ANOVA of SPSS 21.0 (SPSS Inc., Chicago, IL, USA) for chemical composition and fermentation quality. Tukey’s test was used for multiple comparisons and *p* < 0.05 was considered significant. Comparisons of CON vs. LP, CON vs. BC, and LP vs. LP+BC were conducted to evaluate the effects of LP, BC, and their combination on silage fermentation quality and chemical composition using the Kruskal–Wallis test. Comparisons of inoculant treatments were analyzed using t-tests. Differences were considered significant when *p* < 0.05.

## 3. Results

### 3.1. Chemical Composition and Microbial Population of Raw Materials

The chemical composition and microbial populations of fresh materials are presented in Table 1. The pH, WSC, CP, NDF, and ADF contents in wilted alfalfa were 6.49, 19.61 g/kg DM, 201.29 g/kg DM, 476.73 g/kg DM, 311.48 g/kg DM, respectively.

### 3.2. Fermentation Characteristics of Alfalfa Silage

Table 2 shows the fermentation properties of silages. The pH values decreased with prolonged ensiling time. After 3 d of ensiling, the application of LP and BC alone resulted in lower pH values in treated silages compared with the CON silage. However, further application of BC to LP treatment did not lead to any improvement in pH value, with comparable pH values observed in LP and LP+BC-treated silages. On the contrary, LA concentrations increased with prolonged ensiling time, and greater LA concentrations were observed in LP and LP+BC-treated silages. Application of BC did not show significant difference on LA concentration from the CON group regards 3 to 30 days, but increased it after 60 d of fermentation. Compared with LP-treated silages, LP+BC-treated silage had moderately greater LA concentrations at the same ensiling time, significant after 60 d of fermentation. Additionally, the application of LP and BC resulted in decreased AA in treated silages, particularly in LP+BC-treated silages, which being significant lower AA than the CON silage after 14 d of fermentation.

### 3.3. Chemical Composition and Microbial Population of Alfalfa

The chemical composition and microbial population of silages are shown in Table 3. Application of LPor BC had no effects on the contents of DM and DM loss. The treated silages had lower WSC concentration when only LP was applied, while greater WSC concentration was observed when BC was added alone. Compared with LP-treated silage, further application of BC increased WSC concentration in LP+BC-treated silage. Neither LP nor BC affected CP contents but limited proteolysis in treated silages, with lower NH_3_-N concentrations found. In comparison to LP-treated silage, BC-treated silages had comparable NH_3_-N concentrations, while lower NH_3_-N concentration was observed in LP+BC-treated silage when further BC was applied. Both LP and BC decreased NDF and ADF concentrations in the treated silage compared to the CON silage, especially when LP was applied. Moreover, further application resulted in lower ADF in LP+BC-treated silage relative to LP-treated.

The application of LP and BC promoted the growth of LAB in treated silages, especially when their combination was applied. Applying BC had no effects on the population of the yeasts, whereas decreased yeasts were observed in LP-treated silage in comparison to the CON silage. Additionally, molds were not detected in this study.

### 3.4. Microbial Community Composition and Diversity in Alfalfa Silage

The principal coordinates analysis (PCoA) of the beta diversity revealed that the bacterial communities of fresh alfalfa and each treatment group were distinct after fermentation (Figure 1). The first two principal coordinates (PC1 and PC2) accounted for 42.05% and 25.89% of the total variance, respectively. The FM was grouped into a single category, whereas the CON- and BC-treated silages were clustered together and separated from the LP- and LP+BC-treated silages.

The bacterial communities of fresh and silage samples were mainly composed of four phyla (Figure 2A). Before ensiling, Proteobacteria was the most abundant phylum (75.97%), followed by Actinobacteriota (15.53%), Firmicutes (4.98%), and Bacteroidota (2.44%). After fermentation, Firmicutes became the dominant phylum, reaching up to 80.34%, 87.41%, 92.87%, and 88.69% in CON silage, LP-, BC-, and LP+BC-treated silages, respectively. Proteobacteria were more abundant in CON- and LP-treated silages than in BC- and LP+BC-treated silages. At genus level, the predominant genera in FM were *Pseudomonas* (30.50%), *Methylobacterium* (11.48%), *Sphingomonas* (7.91%), *Enterobacter* (7.00%) (Figure 2B). The relative abundance of *Lactobacillus* increased in the treatment group, whereas *Pseudomonas*, *Sphingomonas*, and *Enterobacter* decreased. *Weissella* (41.59%) and *Enterococcus* (27.99%) were the leading genera in CON silage, whereas *Lactobacillus* dominated in LP- and LP+BC-treated silages, reaching 80.73% and 74.98%, respectively. Conversely, BC-treated silages had lower concentrations of *Lactobacillus* (15.28%), but greater concentrations of *Weissella* (42.88%).

Linear discriminant analysis (LDA) effect size (LEfSe) was used to compare different treatments (LDA > 2.0) (Figure 3). There were 13 microbes that differed among the treatments. *Thermoleophilia*, *Stappiaceae* and *Solirubrobacterales* were more abundant in CON silages (LDA > 4.5). The LP-treated silage had higher relative abundances of *Lactobacillus* and *Lactobacillaceae* (LDA > 5.5), whereas the BC-treated silage had greater concentrations of *Enterococcaceae* and *Enterococcus* (LDA > 5.0).

### 3.5. Fermentation Characteristics and Microbial Community Correlation Analysis

The Spearman’s rank correlation between fermentation parameters and microbial strains at genus level is visualized in the form of a heatmap in Figure 4. There was a positive correlation between pH and *Lactococcus*, *Enterococcus*, and *Aerococcus*, but a negative correlation with *Lactobacillus*. The NH_3_-N concentration was positively correlated with the abundance of *norank-f-Solirubrobacteraceae*. Additionally, the LA concentration was positively correlated with *Lactobacillus*, but negatively correlated with *Lactococcus*, *Enterococcus*, *Aerococcus*, *Weissella*, and *Methylobacterium-Methylorubrum*. Conversely, *Lactococcus*, *Brachybacterium*, *Rubellimicrobium*, *Brevundimonas, unclassified-f-Rhizobiaceae, Leuvobacter,* and *unclassified-f-Microbacteriaceae* were negatively correlated with AA concentrations.

### 3.6. Bacterial Metabolic Functions and Enzyme Shifts during Ensiling in Alfalfa Silage

The PICRUSt2 was used to predict the potential functions and enzymes of bacterial communities of the four groups (Figure 5 and Figure 6). The predominant metabolism was the carbohydrate metabolism in silages, followed by amino acid metabolism. Compared to the CON silage, LP and LP+BC- treated silages showed a greater abundance of carbohydrate metabolism and amino acid metabolism, respectively (Figure 5A). The majority of predicted functions explained by KEGG pathways in FM and silages (Figure 5B) were classified into organismal systems (8 pathways), human diseases (12 pathways), environmental information processing (2 pathways), genetic information processing (4 pathways), and metabolism (11 pathways). Compared with FM, alfalfa silages had a higher proportion of amino acid metabolism and lower proportions of carbohydrate metabolism. The carbohydrate metabolism, energy metabolism, cofactors, and vitamin metabolism abundances were lower in treated silages when compared with the CON silage (Figure 5C). The LP+BC-treated silage had the lowest abundances of amino acid metabolism and drug resistance: antimicrobial, the LP-treated and BC-treated silages had comparable abundances. Compared with CON-treated silages, LP-and LP+BC-treated silages had a higher abundance of peptidase, whereas no significant difference was observed between CON- and BC-treated silages (Figure 6). The LP- and LP+BC-treated silages had a lower abundance of cellulose than the control group, and the LP+BC-treated silages had the lowest abundance.

## 4. Discussion

### 4.1. Fermentation Characteristics and Chemical Composition of ALFALFA Silage

It is usually difficult to ensile alfalfa due to its high buffering capacity and low WSC concentration [24], resulting in more significant DM loss in the aerobic phase compared to corn and grass silages [1]. Therefore, reducing the aerobic phase is essential for preserving more nutrients. This study found applying BC resulted in a reduced pH value in treated silage at each sampling time, particularly at 3 d, which is likely due to the acceleration of LA fermentation caused by oxygen consumption. Bai et al. [9,10] also observed similar results when alfalfa silage was treated with *Bacillus* and *Lactobacillus buchneri*. At 60 d of silage, compared to CON silage, further application of BC resulted in lower pH levels and greater concentrations of LA in treated silage. This result confirms the hypothesis and suggests that BC could be an excellent supplement to the current inoculants. After 14 d of ensiling, when LP and BC were used in combination decreased AA concentrations in treated silages. After 60 d of ensiling, a greater LAB population was observed in treated silages than that in the CON silages, which was correlated with the results of lower pH and greater LA concentration. Moreover, lower levels of yeast were found in silages when BC and LP were applied, particularly in LP+BC silage. It appears that both BC and LP used in the study had advantages in nutrition competition over undesirable microbes.

To produce high-quality silage, a WSC is an essential component [25]. The study showed that silages inoculated with LP and its combination with BC had lower WSC concentrations after 60 d of storage; however, greater amounts of WSC were observed in BC-treated silage. It seems that the application of BC could shorten the aerobic phase and preserve more fermentation substrates. For legumes, proteolysis has been identified as a crucial problem in silage fermentation with a greater proportion of non-protein nitrogen, ranging from 440–880 g/kg total nitrogen [26,27]. The proteolysis could result in lower protein utilization efficiency. Therefore, there have been numerous studies evaluating the effects of inoculants on proteolysis, with positive results in legume silage fermentation [28,29]. The concentration of NH_3_-N, which is an indicator of proteolysis [30,31], was found to be lower in LP and BC-treated silages when compared to the CON silage, particularly when both BC and LP were applied. This could be attributed to the lower pH levels in treated silages. Additionally, lower NDF and ADF concentrations were found in treated silages when LP and BC were applied. These results could be explained by the acid degradation of hemicellulose. Furthermore, *Bacillus* could produce enzymes such as cellulase and feruloyl esterase, and it has been demonstrated that BC could degrade the lignin-related compounds ferulic acid and vanillin to vanillic acid [11]. This could be the reason for lower ADF concentrations in LP+BC-treated silage when compared with silage inoculated with LP alone.

### 4.2. Diversity and Composition of Microbial Community

Silage fermentation involves the interactions of microbes, and a better understanding of the dynamic of microbial communities could be beneficial for regulating the fermentation process [32]. In this study, we investigated the effects of LP and BC on the bacterial communities of ensiled alfalfa. The results showed that fresh alfalfa was dominated by Proteobacteria and Firmicutes, which is in agreement with previous findings [33]. Li et al. [34] reported similar results for Cassava foliage after 60 days of ensiling with an increase in the relative abundance of Firmicutes to 88.69%.

To further evaluate the effects of BC on the bacterial community during fermentation, we also analyzed the bacterial structures of alfalfa at the genus level. The dominant bacteria in the fresh alfalfa were Gram-negative bacteria, including *Pseudomonas*, *Methylobacterium-Methylorubrum*, and *Sphingomonas*. According to a previous report, the microbial flora of the alfalfa phyllosphere was predominately composed of *Erwinia*, *Escherichia*, *Pseudomonas*, and *Pantoea* [35]. This might be because the settling of bacteria on the plant surface is affected by plant species, climate, geographical location, and fertilizer type [32]. *Lactobacillus* and *Weissella* are the most common bacteria involved in the LA fermentation of silage [36,37]. The researchers discovered that, as compared to the CON silage, treated silages showed greater relative abundances of *Lactobacillus* after 60 d, especially in LP and LP+BC-treated silages. This could be attributed to the lower pH and greater lactic acid production. Similar results were reported by Bai et al. [9,10], who found that the abundance of *Lactobacillus* was higher in *Bacillus subtilis* and *Bacillus amyloliquefaciens*-treated silages than in the CON silage. *Weissella*, which was considered an early colonizer, had a lower acid tolerance, and was unable to survive when the pH was below 4.5, yet the relative abundance of *Weissella* in BC-treated silage increased from 41.59% to 42.88% after 60 d of ensiling [38,39]. This suggested that BC was not as effective in competing for nutrients with *Weissella*.

### 4.3. Relationships between Fermentation Parameters

A correlation analysis was conducted between the composition of microorganisms and end-products after ensiling. It was observed that *Methylobacterium-Methylorubrum*, an aerobic bacterium utilizing the serine pathway to consume methanol and other reduced one-carbon compounds [18], was gradually replaced by *Lactobacillus* in this study as the pH of the silage decreased, resulting in a positive correlation between pH and *Methylobacterium-Methylorubrum* and a negative correlation between pH and *Lactobacillus*. *Lactobacillus* is essential in inhibiting harmful microorganisms by rapidly acidifying silage in the late stages of ensiling [40]. In this study, a positive correlation was observed between NH_3_-N concentration and *norank-f-Solirubrobacteraceae*. This is likely due to the ability of certain species of *Solirubrobacteraceae* to thrive in anaerobic conditions and competes with *Lactobacillus* for nutrients [41]. These results also agreed with the greater NH_3_-N concentration and relative abundance of *norank-f-Solirubrobacteraceae* in CON silage. Additionally, LA and AA concentrations were shown to be positively connected with *Lactobacillus* and negatively correlated with *Enterobacter*, which is likely a result of the production of LA and AA by *Lactobacillus*. These findings were further supported by the higher NH_3_-N concentration and relative abundance of *norank-f-Solirubrobacteraceae* in CON silage.

### 4.4. Bacterial Metabolic Functions and Enzyme Shifts in the Silage

Alfalfa silage bacterial community functions were predicted using the KEGG pathways database and PICRUSt2. All samples contained a high proportion of the KEGG metabolism pathways, including the carbohydrate metabolism and amino acid metabolism. It was observed that four metabolic categories had a significant impact on microbial metabolisms. These categories were secondary metabolite biosynthesis, microbial metabolism in various environments, carbon metabolism, 2-oxocarboxylic acid metabolism, fatty acid metabolism, amino acid biosynthesis, and aromatic compound degradation [42]. After ensiling, microbial metabolism was inhibited in the anaerobic environment, resulting in weaker amino acid metabolism. However, LP-treated silage had a greater abundance of *Lactobacillus* than LP+BC-treated silages, which contributed to a higher microbial metabolism capacity.

We concentrated our efforts on the metabolic pathways of carbohydrates, amino acids, energy, cofactors, vitamins, and drug resistance, as these are all associated with changes in fermentation, chemical composition, and human health [43,44]. Du et al. [18] determined that the relative abundance of total LAB in the microbial community affected the abundance of carbohydrates metabolism pathways. However, this study found that entire LAB was lower in silage with more carbohydrate metabolic pathways. Because carbohydrate metabolism is primarily glycolysis and gluconeogenesis, *Enterococcus* are more capable of metabolizing WSC in CON silage and LP+BC-treated silages than beneficial species of LAB [45]. Amino acids are integral to proteins and peptides, and play a substantial role in the energy metabolism and environmental tolerance of lactic acid bacteria [46]. The application of LP, BC, and their combination resulted in lower amino acid metabolism abundances, which correlates with the reduced NH_3_-N concentrations observed in the treated silages compared to the CON silage.

The relative abundance of cofactor and vitamin metabolism was higher in treated silages, suggesting that BC could accelerate vitamin production or produce vitamins directly during ensiling. Similar results were reported by Bai et al. [47] who found that inoculating alfalfa silage with *Enterococcus faecalis* enhanced the relative abundance of cofactor and vitamin metabolism. Abuse of antibiotics leads to the development of multidrug-resistant bacteria, one of the greatest health threats [48,49]. The antimicrobial drug resistance relative abundance was found to be the greatest in the CON silage, whereas the LP+BC-treated silage had the lowest relative abundance, followed by LP-treated silages. Antimicrobial drug resistance is primarily found in undesirable bacteria, such as *Salmonella* and *Escherichia coli* [44]. The inhibition of undesirable and hazardous bacteria may be related to the lower pH in LP+BC-treated silages, which reduces resistance genes.

## 5. Conclusions

The addition of BC increased the fermentation quality of alfalfa silages, indicated by lower pH values and greater LA concentrations, especially when BC was applied together with LP. Furthermore, the application of BC decreased DM loss, NDF, ADF and NH_3_-N contents, while increasing WSC. The microbial analysis demonstrated that the application of BC increasing *Lactobacillus* abundance and decreasing *Enterococcus* abundance in treated silages after 60 d of fermentation. Additionally, the application of LP, BC, and their combinations increased cofactor and vitamin metabolism abundance while decreasing drug resistance: antimicrobial pathway abundance. Thus, BC could be considered a viable bioresource for improving fermentation quality.

## Figures and Tables

**Figure 1 animals-13-00932-f001:**
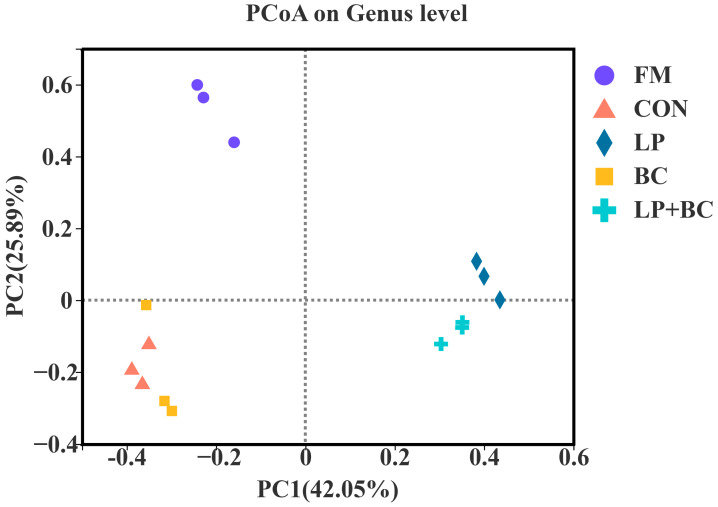
Bacteria community principal coordinates analysis (PCoA) on genus level for alfalfa silage. FM, fresh shrub; CON, control silage; LP, silages inoculated with *Lactobacillus plantarum*; BC, silages inoculated with *Bacillus coagulans*; and LP+BC, silages combined with *Lactobacillus plantarum* and *Bacillus coagulans*.

**Figure 2 animals-13-00932-f002:**
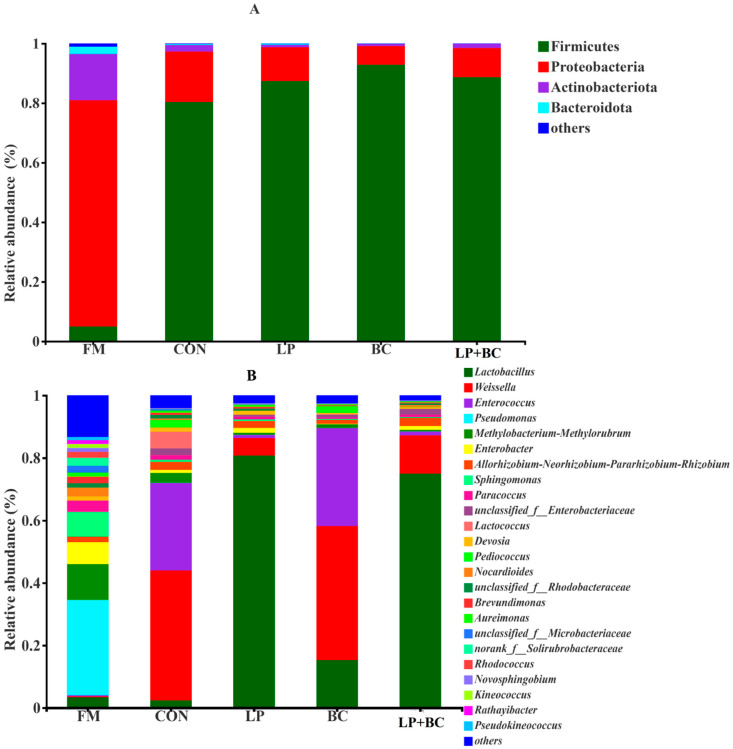
Bacterial communities and relative abundance by phylum level (**A**) and genus level (**B**) for alfalfa silage after 60 d ensiling. FM, Fresh material; CON, control silage; LP, silages inoculated with *Lactobacillus plantarum*; BC, silages inoculated with *Bacillus coagulans*; and LP+BC, silages combined with *Lactobacillus plantarum* and *Bacillus coagulans*.

**Figure 3 animals-13-00932-f003:**
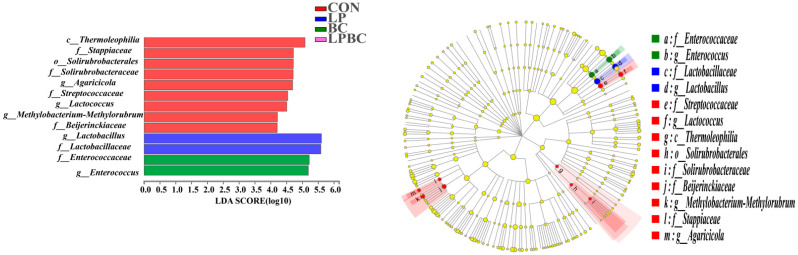
Comparison of microbial variations after 60 d of ensiling using the LEfSe analysis. CON, control silage; LP, silages inoculated with *Lactobacillus plantarum*; BC, silages inoculated with *Bacillus coagulans*; and LP+BC, silages combined with *Lactobacillus plantarum* and *Bacillus coagulans*.

**Figure 4 animals-13-00932-f004:**
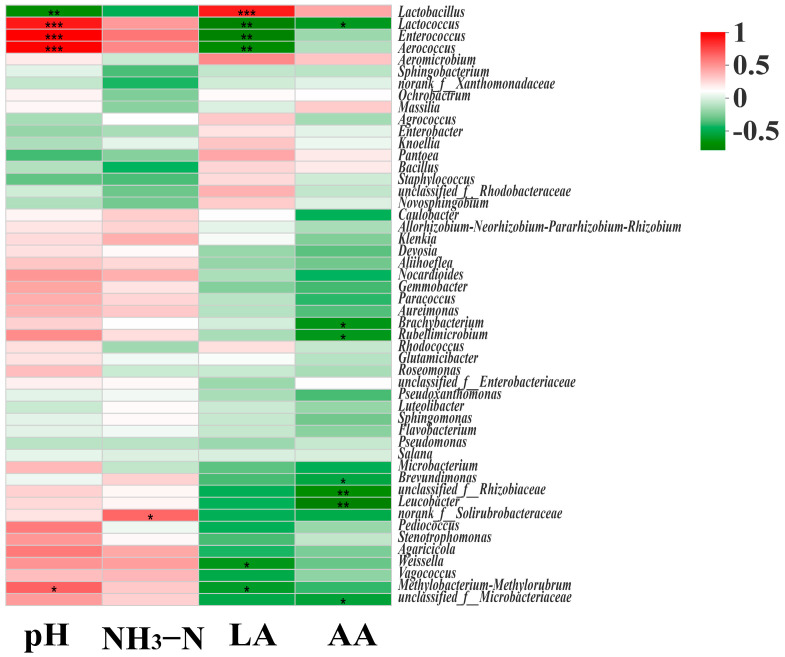
Correlation analysis between bacterial community and fermentation characteristics of alfalfa. LA, lactic acid; AA, acetic acid; NH_3_-N, ammonia nitrogen. * represents *p* < 0.05, ** represents *p* < 0.01, and *** represents *p* < 0.001. Different color ranges represent different correlation coefficients in the right legend.

**Figure 5 animals-13-00932-f005:**
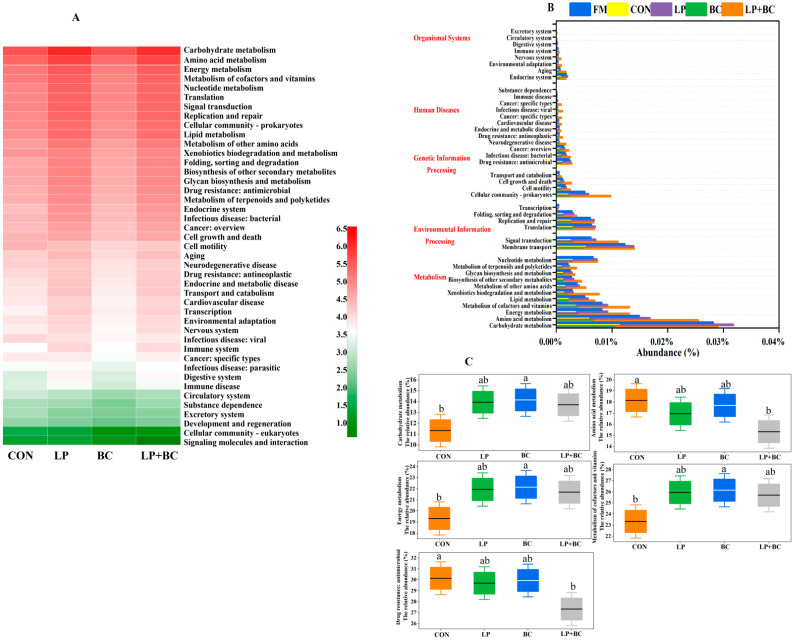
Bacterial alterations that contribute to functional shifts after fermentation in different groups. CON, control silage; LP, silages inoculated with *Lactobacillus plantarum*; BC, silages inoculated with *Bacillus coagulans*; and LP+BC, silages combined with *Lactobacillus plantarum* and *Bacillus coagulans*. The Phylogenetic Investigation of Communities by Reconstruction of Unobserved States (PICRUSt2) predicts functional shifts. The second level of the predicted functional shift for each Kyoto Encyclopedia of Genes and Genomes (KEGG) pathway is shown for each group. (**A**). Alfalfa functional abundance (top 50 abundant functions) before and after ensiling. In the middle heat map, the color corresponds to the Z value calculated after normalizing the relative abundance of the function. The closer the color is to red, the greater the abundance. (**B**). Level 2 KEGG orthologue functional predictions explained by PICRUSt2. (**C**). Significant differences in the 5 functional pathways among the four groups at *p* < 0.05. Different lowercase letters shows difference among treatments (*p* < 0.05).

**Figure 6 animals-13-00932-f006:**
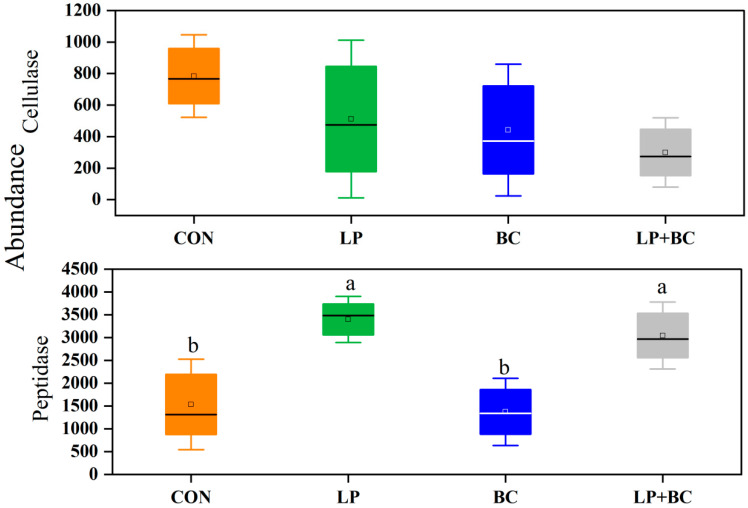
Bacterial alterations that contribute to enzyme shifts after fermentation in different groups. CON, control silage; LP, silages inoculated with *Lactobacillus plantarum*; BC, silages inoculated with *Bacillus coagulans*; and LP+BC, silages combined with *Lactobacillus plantarum* and *Bacillus coagulans*. Different lowercase letters indicate differences between treatments in cellulase and protease (*p* < 0.05).

**Table 1 animals-13-00932-t001:** Chemical composition and microbial population of alfalfa before ensiling.

Item ^1^	Alfalfa
DM, g/kg FW	329.60
pH	6.49
WSC, g/kg DM	19.61
CP, g/kg DM	201.29
EE, g/kg DM	14.75
NDF, g/kg DM	476.73
ADF, g/kg DM	311.48
LAB, log_10_ CFU/g FW	6.41
Yeasts, log_10_ CFU/g FW	3.28
Molds, log_10_ CFU/g FW	2.26

^1^ DM, dry matter; WSC, water-soluble carbohydrate; CP, crude protein; NDF, neutral detergent fiber; ADF, acid detergent fiber; EE, ether extract; CFU, colony-forming unit; LAB, lactic acid bacteria; FM, fresh matter.

**Table 2 animals-13-00932-t002:** Fermentation characteristics of the alfalfa silage during ensiling.

Items ^1^	Ensilage Time, d	Treatments ^2^				SEM ^3^	*p*-Value			
		CON	LP	BC	LP+BC		CON vs. LP	CON vs. BC	LP vs. LP+BC	I ^4^
pH	3	6.13 ^Aa^	5.42 ^Ac^	5.79 ^Aa^	5.36 ^Ac^	0.04	<0.001	0.034	0.84	<0.001
	7	6.04 ^Aa^	5.40 ^Ac^	5.80 ^Ab^	5.33 ^Ac^	0.04	<0.001	<0.001	0.31	<0.001
	14	5.83 ^Ba^	5.18 ^Bc^	5.68 ^Ab^	5.17 ^Bc^	0.03	<0.001	0.05	0.90	<0.001
	30	5.69 ^Ba^	4.96 ^Cc^	5.27 ^Bb^	4.94 ^Cc^	0.03	<0.001	<0.001	0.54	<0.001
	60	5.38 ^Ca^	4.89 ^Cc^	5.14 ^Cb^	4.85 ^Cc^	0.04	<0.001	<0.001	0.57	<0.001
LA, g/kg DM	3	15.41 ^Db^	25.45 ^Ca^	18.51 ^Cb^	26.48 ^Ca^	0.12	0.001	0.24	0.64	<0.001
	7	31.92 ^Cc^	59.01 ^Bb^	58.23 ^Bb^	63.84 ^Ba^	0.54	<0.001	<0.001	0.69	0.004
	14	52.87 ^Bb^	79.48 ^Aa^	57.03 ^Bb^	85.06 ^ABa^	0.46	0.01	0.30	0.54	0.001
	30	67.06 ^ABc^	87.25 ^Aab^	72.64 ^Abc^	90.66 ^Aa^	0.21	0.02	0.46	0.57	0.003
	60	74.88 ^Ac^	91.55 ^Aab^	82.12 ^Ab^	93.83 ^Aa^	0.19	0.02	0.05	0.04	<0.001
AA, g/kg DM	3	20.28 ^C^	9.80 ^C^	10.78 ^C^	8.05 ^B^	0.30	0.02	0.09	0.67	0.06
	7	24.88 ^BC^	17.96 ^BC^	23.23 ^BC^	14.60 ^AB^	0.29	0.21	0.74	0.31	0.09
	14	35.50 ^ABa^	27.09 ^Bab^	32.26 ^ABa^	16.82 ^ABb^	0.29	0.03	0.56	0.01	0.004
	30	47.32 ^Aa^	38.89 ^Aa^	45.03 ^Aa^	19.10 ^ABb^	0.32	0.09	0.53	0.013	<0.001
	60	47.19 ^Aa^	42.19 ^Aa^	43.59 ^Aa^	23.87 ^Ab^	0.29	0.34	0.49	<0.001	<0.001

Means with different uppercase superscripts in the same column differ significantly (*p* < 0.05); Means with different lowercase superscripts in the same row differ significantly (*p* < 0.05). ^1^ LA, lactic acid; AA, acetic acid; DM, dry matter; ^2^ CON, control silage; LP, silage inoculated with *Lactobacillus plantarum*, BC, silage inoculated with *Bacillus coagulans*; LP+BC, silage inoculated with the conbination of *Lactobacillus plantarum* and *Bacillus coagulans*; ^3^ SEM, error of the means; ^4^ I, inoculants treatment.

**Table 3 animals-13-00932-t003:** Chemical compositions and microbial population of the alfalfa after 60 d of ensiling.

Item ^1^	Treatments ^2^				SEM ^3^	*p*-Value			
	CON	LP	BC	LP+BC		CON vs. LP	CON vs. BC	LP vs. LP+BC	I ^4^
DM, g/kg	432.47	388.31	367.11	394.80	0.18	0.26	0.13	0.51	0.17
DM loss, %	10.61	8.25	9.65	7.98	1.45	0.52	0.84	0.95	0.93
WSC, g/kg	12.88	7.93	14.15	12.10	0.06	0.01	0.33	0.01	0.001
CP, g/kg DM	208.34	211.09	210.45	216.65	3.54	0.18	0.69	0.31	0.42
NDF, g/kg DM	391.29	363.00	374.00	361.78	0.18	0.01	0.10	0.11	<0.001
ADF, g/kg DM	269.53	247.00	254.00	214.83	0.14	0.004	0.02	<0.001	<0.001
NH_3_-N, g/kg total N	142.00	126.51	130.91	110.90	256	0.003	0.03	0.004	<0.001
LAB, log_10_ CFU/g	7.41	7.71	7.61	7.91	0.02	<0.001	<0.001	<0.001	<0.001
Yeasts, log_10_ CFU/g	2.79	2.62	2.71	2.53	0.02	0.002	0.06	0.01	<0.001
Molds, log_10_ CFU/g	<2.0	<2.0	<2.0	<2.0	-				

^1^ DM, dry matter; DM loss, dry matter loss; WSC, water-soluble carbohydrate; CP, crude protein; NDF, neutral detergent fiber; ADF, acid detergent fiber; NH_3_-N, ammonia nitrogen; CFU, colony-forming unit; LAB, lactic acid bacteria; ^2^ CON, control silage; LP, silage inoculated with *Lactobacillus plantarum*, BC, silage inoculated with *Bacillus coagulans*; LP+BC, silage inoculated with the conbination of *Lactobacillus plantarum* and *Bacillus coagulans*; ^3^ SEM, error of the means; ^4^ I, inoculants treatment.

## Data Availability

All the sequence data of this study were deposited in the NCBI Sequence Read Archive (SRA) database under the accession number PRJNA935541.

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
