# Peer review of "Effects of Bacillus coagulans and Lactobacillus plantarum on the Fermentation Characteristics, Microbial Community, and Functional Shifts during Alfalfa Silage Fermentation"

_animals, 2023, doi:10.3390/ani13050932_

Round 1

Reviewer 1 Report

Comments to the Authors,

This manuscript investigated the potential of Bacillus coagulans (BC) as an inoculant in silage fermentation by comparing its effects with Lactobacillus plantarum on silage fermentation quality, bacterial community dynamics and functional shifts. Few literatures have evaluated the effects of aerobic strains on silage fermentation. This study was based on dry matter loss in aerobic stage and aimed to reduce nutrient loss by using an aerobic strain to consume oxygen and reduce respiratory loss. It is a certain amount of innovation and could be something interesting. However, there are still some problems need to be solved. The figures used are not clear enough and need further improvements. In addition, there are some spelling and grammar mistakes, the authors should check. Some grammar mistakes are shown in below:

2 Correct “Lactobacillus Plantarum to “Lactobacillus plantarum”

9 Chang “each stage of production” to “silage production”

12-13 Change “beneficial anaerobic intestinal bacteria” to “anaerobic bacteria”, same in line 59

16 It should be “the contents of neutral detergent fiber”, please check.

25 The application rate of each strain should be added in LP+BC treatment.

29 Chang “mainly” to “especially”

30 Delete concentration

33 Change “the control” to the CON

59 Change “it” to “which”

61 Change “are” to “have been”

66 Correct “fish” to “fishes

83 How long did it take to wilt the fresh alfalfa?

92 Correct “All silage is” to “All silages were”

98 Change “milling them through a 1-mm screen” to “ground with a mill (1-mm screen).”

114 Delete “21.”

167 “being significant”

221-222 This result was not that correct according to Figure 2. Compared with FM, CON silage had lower Lactobacillus, please check.

263-264 It should be “alfalfa silages had higher proportions of”

269 In Figure 5C and 6, we should use different letters to show the differences among treatments.

324 “involves” not “iinvloves”

409 The final conclusion still needs some improvements, and it should be focus on the topic whether BC could be applied in silage fermentation.

Author Response

Introductory remark

In the revised version prepared, all changes made in text are highlighted by colored text. The comments and corrections suggested by the referees were completely adopted unless the corresponding text had been otherwise modified. In the revised version, the red colored words were the responses to the comments of Reviewer #1; the blue colored words were the responses to the comments of Reviewer #2, the yellow colored words were the responses to the comments of Reviewer #3, the green colored words were the responses to the comments of Reviewer #4, the highlighted words were corrected by a native English speaker. We highly appreciate the constructive and kind comments that all the reviewers addressed on our manuscript.

Reviewer #1 comments and our responses

  1. Line -2 Correct “Lactobacillus Plantarumto “Lactobacillus plantarum

Response: It has been corrected.

  1. Line -9 Chang “each stage of production” to “silage production”

Response: It has been corrected.

  1. Line -12-13 Change “beneficial anaerobic intestinal bacteria” to “anaerobic bacteria”, same in line 59

Response: It has been corrected.

  1. Line -16 It should be “the contents of neutral detergent fiber”, please check.

Response: We have checked the manuscript throughly and corrected all wrong description, thank you for your kind suggestion.

  1. Line -25 Application rates for LP+BC treatments have been supplemented.

Response: We have supplemented: “BC and LP were applied to the LP+BC-treated group (1 × 106 CFU/g FW, respectively)”.

  1. Line -29 Chang “mainly” to “especially”

Response: It has been corrected.

  1. Line -30 Delete concentration

Response: It has been deleted.

  1. Line -33 Change “the control” to“the CON”

Response: It has been corrected.

  1. Line -59 Change “it” to “which”

Response: It has been corrected.

  1. Line -61 Change “are”to “have been”

Response:It has been corrected.

  1. Line -66 Correct “fish” to “fishes”

Response: It has been corrected.

  1. Line-83 How long did it take to wilt the fresh alfalfa?

Response: within 8 h.

  1. Line -92 Correct “All silage is” to “All silages were”

Response: It has been corrected.

  1. Line -98 Change “milling them through a 1-mm screen” to “ground with a mill (1-mm screen).”

Response:It has been corrected.

  1. Line-114 Delete the “21”.

Response: This is a reference and has been corrected.

  1. Line-167 “being significant”

Response: It has been corrected.

  1. Line-221-222 This result was not that correct according to Figure 2. Compared with FM, CON silage had lower Lactobacillus, please check.

Response: We have corrected as “The relative abundance of Lactobacillus increased in the treatment group, while Pseudomonas, Sphingomonas, and Enterobacter decreased”.

  1. Line-263-264 It should be “alfalfa silages had higher proportions of”

Response: It has been corrected.

  1. Line-269 In Figure 5C and 6, we should use different letters to show the differences among treatments.

Response: It has been corrected.

  1. Line-324 “involves”not “iinvloves”

Response: It has been corrected.

  1. Line-409 The final conclusion still needs some improvements, and it should be focus on the topic whether BC could be applied in silage fermentation.

Response: We have supplied as “ Furthermore, the application of BC decreased NDF, ADF and NH3-N contents, while increasing WSC”.

Author Response

Introductory remark

In the revised version prepared, all changes made in text are highlighted by colored text. The comments and corrections suggested by the referees were completely adopted unless the corresponding text had been otherwise modified. In the revised version, the red colored words were the responses to the comments of Reviewer #1; the blue colored words were the responses to the comments of Reviewer #2, the yellow colored words were the responses to the comments of Reviewer #3, the green colored words were the responses to the comments of Reviewer #4, the highlighted words were corrected by a native English speaker. We highly appreciate the constructive and kind comments that all the reviewers addressed on our manuscript.

Reviewer #2 comments and our responses

  1. Bacillus coagulans should be written consistently throughout the text, with both "Bacillus coagulans" and "Bacillus Coagulans" occurring.

Response: It has been corrected to “Bacillus coagulans”.

  1. Figure 3 is not clear enough.

Response: Have improved the clarity of Figure 3.

  1. In Table 3, microbial count units in accordance with Table 1.

Response: Table 3 Microbial count units have been harmonized with Table 1

  1. Line 16 “The contents of neutral detergent fiber”should be used instead.

Response: It has been corrected.

  1. Line 23 Dry matter (DM) content of 329 g/kg should be harmonized with the results.

Response: we have corrected as “329.60 g/kg”

  1. Line 83 dry matter (DM) content of 329 g/kgshould be consistent with Table 1.

Response: we have corrected as “329.60 g/kg”

  1. Line 92 substitute “all silages were” for “all silage is”.

Response: It has been deleted.

  1. Line 92 change “CON” to “the CON”.

Response: It has been corrected.

  1. Line 170 Delete “Propionic acid”.

Response: It has been deleted.

  1. Line 342 change“abundances ” to “abundance”.

Response: It has been corrected.

  1. Line 344-345 change“ Weissella, which was considered an early colonizer, had a lower acid tolerance and unable to survive when the pH was below 4.5”, to “ Weissella, which was considered an early colonizer, had a lower acid tolerance, and was unable to survive when the pH was below 4.5”.

Response: It has been corrected.

  1. Conclusion needs to be improved, for example, the effect of supplementation on NH-N3and DM loss.

Response: We have supplied as“ Furthermore, the application of BC decreased NDF, ADF and NH3-N contents, while increasing WSC”.

BC has no significant effect on DM loss and therefore is not described

13.The authors are advised to check the format of the cited references. For instance, lines 474-454 "[17] “Liu, Q.; Bao, X.; Guo, G.; Huo, W.; Xu, Q.; Wang, C.; Li, Q.; Liu, Q. Effects of hydrolysable tannin with or without condensed tannin on alfalfa silage fermentation characteristics and in vitro ruminal methane production, fermentation patterns, and microbiota. Anim. 2021, 11, 1967, ” author sorting error.

Response: thank you very much for your kind suggestion. I have checked the references, and corrected author order of [17].

Reviewer 3 Report

This is a thesis on the utilization of microbial additives for the preparation of good alfalfa silage. Freshness using a new type of microorganisms stands out. Please correct the details.

Author Response

Introductory remark

In the revised version prepared, all changes made in text are highlighted by colored text. The comments and corrections suggested by the referees were completely adopted unless the corresponding text had been otherwise modified. In the revised version, the red colored words were the responses to the comments of Reviewer #1; the blue colored words were the responses to the comments of Reviewer #2, the yellow colored words were the responses to the comments of Reviewer #3, the green colored words were the responses to the comments of Reviewer #4, the highlighted words were corrected by a native English speaker. We highly appreciate the constructive and kind comments that all the reviewers addressed on our manuscript.

Reviewer #3 comments and our responses

  1. Line -16: Delete “content of” and NDF= neutral detergent fiber

Response: It has been corrected.

  1. Line -23: “alfalfa” should be added before “silage fermentation”

Response: It has been corrected.

  1. Line -29-31: Suggesting simplification. Likely “application of BC” and “BC-treated silage” repeated

Response: We have corrected: “Application of BC preserved more water-soluble carbohydrates (WSC), and further application of BC increased WSC in LP+BC-treated silage compared to LP-treated silage”.

  1. Line -32: “control” → “CON”

Response: It has been corrected.

  1. Line -36-37: “in treated silage” should be deleted.

Response: It has been deleted.

  1. Line -48: “quality of silage” → “silage quality”

Response: It has been corrected.

  1. Line -83: “a” → “the”

Response: It has been corrected.

  1. Line -86-87: Is there no replication for each treatment? Why is just 20? If have

replications, it should be 5*4*3. Please check it.

Response: We have corrected: “The remaining 60 piles (5 ensiling durations × 4 treatments × 3 replications)”.

  1. Line -114: What is “21”? I think it may be a reference, please check it again.

Response: This is a reference and has been corrected

  1. Line -142: “were” → “was”

Response: It has been corrected.

  1. Line -158-160: The pH values are not stable but have a downward trend from day 30 to 60. Please check the data again.

Response: We have corrected: “The pH values decreased with prolonged ensiling time”.

  1. Why is the lactic acid concentration of LP+BC significantly higher than LP but

there is no significant difference in the pH values after 60 days?

Response: Because the LA produced is not enough to lower the pH value.

  1. BC had no significant effect on the inhibition of acetic acid production compared with the CON (p>0.05). But why can it significantly inhibit the production of acetic acid when it is used in combination with LP from day 14 to 60?

Response: In this study, BC had no significant effect on acetic acid concentration, but when mixed with LP, it could shorten the aerobic stage of early fermentation and make it more suitable for LP growth. The growth of LP obviously inhibits the production of acetic acid.

  1. Line -192-193: LP can significantly reduce yeasts compared with CON (p=0.002<0.05). Please check the result analysis.

Response: We have corrected: “Applying BC had no effects on the population of the yeasts, while decreased yeasts were observed in LP-treated silages in comparison to the CON silages”.

  1. Why is a large amount of NH3-N (>110 g/kg) produced but CP content increases after ensiling?

Response: The crude protein content during silage represents the total nitrogen content and is associated with dry matter loss. In this study, both dry matter loss and crude protein content in the treatment group were numerically lower than those in the control group. However, the NH3-N only represented the degree of protein degradation, and the treatment group inhibited protein degradation by rapidly reducing pH. Thus, there is no contradiction between a decrease in NH3-N and a slight increase in crude protein. Bai et al (2020) had also observed similar results that large amount of NH3-N produced but CP content increases after ensiling.

Bai, J.; Xu, D.; Xie, D.; Wang, M.; Li, Z.; Guo, X. Effects of antibacterial peptide-producing Bacillus subtilis and Lactobacillus buchneri on fermentation, aerobic stability, and microbial community of alfalfa silage. Bioresour. Technol. 2020, 315, 123881.

  1. Why is the data of LP+BC not shown in Figure 3?

Response: Figure 3 is based on the LEfse results, and the LEfse results did not show LP+BC.

  1. Line -263: “Metabolism”→ “metabolism”

Response: It has been corrected.

18: Line -295-296: pH or LA concentration (p>0.05) when LP vs LP+BC

Response: We have corrected: “At 60 d of silage, compared to CON silage, further application of BC resulted in lower pH and greater LA concentration in treated silage”.

19: Line -298-299: When LP or BC has applied alone, the acetic acid content did not decrease significantly compared with the CON (p>0.05).

Response: We have corrected: “After 14 d of ensiling, when LP and BC were used in combination decreased AA concentrations in treated silages”.

20: Line -302-303: “the molds were not detected in this study” in lines 193-194 was reported by the author. It’s contradictory, and from table 3, we can see that molds <2 log CFU/g in all groups. Please check

Response: We are very sorry that this refers to yeasts. It has been corrected.

  1. Line -324: ‘involves” spelling mistake, please check

Response: It has been corrected.

Reviewer 4 Report

The present paper, titled

 “Effects of Bacillus coagulans and Lactobacillus Plantarum on 2 the fermentation characteristics, microbial community, and 3 functional shifts during alfalfa silage fermentation” assessed the effects of the inoculation of Bacillus coagulans with or without Lactobacillus Plantarum on alfalfa silage quality and microbial community.

The topic is interesting and adds knowledge on the dynamics occurring during ensiling.

I have some concerns as regards the description of material and methods that is not complete, especially regarding the statistical analysis used.

In the results and discussion chapters the effect of BC inoculation on silage quality is not always consistent with the results presented in the Tables.

For these reasons I recommend minor revisions.

Specific comments

L16: neutral detergent fiber

L142: Please, improve the description of statistical analysis. Specify what data were analysed through one-way ANOVA ad Kruskal-Wallis Test. What fixed effects did you consider? Were data analysed through ANOVA normally distributed? What test did you use to assess normal distribution? Did you consider the effect of ensilage time?

No descriptions of PCoa, L LEfSe, Spearman’s rank correlation and PICRUSt2 are presented. Please add them.

L149-151: Avoid to repeat the results in the text if they are already presented in the tables.

L 158-160 and Table 2: I don’t see the significance of Ensilage time

L161-162: please, change into “further application of BC to LP treatment did not…

L163-165: Please be more specific and strictly follow the results presented in Table 2. BC is not significant as regards LA and LP+BC is not significantly different from CON as regards 3 to 30 days.

L171: Propionic acid is not reported in the table.

L 177-188: Here and along the whole manuscript if the p value is >0.1 there is not a trend of significance, but the difference is just not significant. Please, do not write that the values are higher or lower if there is not significance, just write not significant.

L257: the meaning of PICRUSt2 is not defined. Please add the explanation of the abbreviations at their first use.

L258,-259: please explain better.

L260, 275: please, explain KEGG

L271-280: The caption of Figure 5 is not complete. The description of Letter C is missing.

Figure 5: The p values of the correlation and of the differences presented in this figure are not clear. Please specify it. The resolution of the figure must be improved, since it is difficult to read it, especially the part B.

L294-298: according to Table 2 these results are not consistently significant. Please rephrase.

L298-299: BC alone did not significantly reduced AA, see Table 2. Please rephrase.

L302-303: do you mean yeasts? Since molds were not found (Table 3)

Author Response

Introductory remark

In the revised version prepared, all changes made in text are highlighted by colored text. The comments and corrections suggested by the referees were completely adopted unless the corresponding text had been otherwise modified. In the revised version, the red colored words were the responses to the comments of Reviewer #1; the blue colored words were the responses to the comments of Reviewer #2, the yellow colored words were the responses to the comments of Reviewer #3, the green colored words were the responses to the comments of Reviewer #4, the highlighted words were corrected by a native English speaker. We highly appreciate the constructive and kind comments that all the reviewers addressed on our manuscript.

Reviewer #4 comments and our responses

  1. Line-16: neutral detergent fiber

Response: It has been corrected.

  1. Line-142: Please, improve the description of statistical analysis. Specify what data were analysed through one-way ANOVA ad Kruskal-Wallis Test. What fixed effects did you consider? Were data analysed through ANOVA normally distributed? What test did you use to assess normal distribution? Did you consider the effect of ensilage time?

No descriptions of PCoa, LEfSe, Spearman’s rank correlation and PICRUSt2 are presented. Please add them.

Response: We have supplemented: “Principal coordinates analysis (PCoA) was drawn using the Vegan and Python packages. The linear discriminant analysis effect size (LEfSe) was used to calculate the communities or species with significant differences among the four groups. To investigate the relationship between silage quality and the bacterial community, Spearman’s rank correlation coefficients were generated using R software (Microgen). Using the relative abundances of microbes at the species level, the top 50 independent variables were calculated, and the dependent variables were calculated based on pH, NH3-N, LA, and AA. The bacterial function was predicted using the Kyoto Encyclopedia of Genes and Genomes (KEGG) database Phylogenetic Survey of Communities in Unobserved States Reconstruction (PICRUSt2, v2.3.0_b, https://github.com/picrust/picrust2), which predicts the functional abundance of a sample based on the abundance of tagged gene sequences in the samples”.

Data was analyzed using One-Way ANOVA of SPSS 21.0 (SPSS Inc., Chicago, IL, USA) for chemical composition and fermentation quality. Tukey’s test was used for multiple comparisons and p < 0.05 was considered significant. Comparisons of CON vs LP, CON vs BC, and LP vs LP+BC were conducted to evaluate the effects of LP, BC, and their combination on silage fermentation quality and chemical composition using the Kruskal-Wallis test. Comparisons inoculants treatment were analyzed using t-tests. Differences were considered significant when p < 0.05.

  1. Line-149-151: Avoid to repeat the results in the text if they are already presented in the tables.

Response: It has been corrected, thank you.

  1. Line-158-160 and Table 2: I don’t see the significance of Ensilage time.

Response: Significance analysis has been done for different fermentation times in Table 2.

  1. Line-161-162: please, change into “further application of BC to LP treatment did not”

Response: It has been corrected.

  1. Line-163-165: Please be more specific and strictly follow the results presented in Table 2. BC is not significant as regards LA and LP+BC is not significantly different from CON as regards 3 to 30 days.

Response: It has been corrected, thank you.

  1. Line-171: Propionic acid is not reported in the table.

Response: It has been deleted.

  1. Line-177-188: Here and along the whole manuscript if the p value is >0.1 there is not a trend of significance, but the difference is just not significant. Please, do not write that the values are higher or lower if there is not significance, just write not significant.

Response: We have corrected: “Application of LP or BC had no effects on the contents of DM and DM loss”. 

  1. Line-257: the meaning of PICRUSt2 is not defined. Please add the explanation of the abbreviations at their first use.

Response: Line-257, PICRUSt2 is explained in the material and methods.

  1. Line-258,-259: please explain better.

Response: We have supplemented: “The predominant metabolism was the carbohydrate metabolism in silages, followed by amino acid metabolism. Compared to the CON silage, LP and LP+BC- treated silages showed a greater abundance of carbohydrate metabolism and amino acid metabolism respectively (Figure 5A)”.

  1. Line-260, 275: please, explain KEGG

Response: Line-260, KEGG is explained in the material and methods.

Line-275, KEGG explanations have been added.

  1. Line-271-280: The caption of Figure 5 is not complete. The description of Letter C is missing.

Response: We have supplemented: “C. Significant differences of the 5 functional pathways among the four groups at p < 0.05”.

  1. Figure 5: The p values of the correlation and of the differences presented in this figure are not clear. Please specify it. The resolution of the figure must be improved, since it is difficult to read it, especially the part B.

Response: Figure 5 we have used different letters to show the differences among treatments and increase the resolution of the image.

  1. Line-294-298: according to Table 2 these results are not consistently significant. Please rephrase.

Response: We have corrected: “At 60 d of silage, compared to CON silage, further application of BC resulted in lower pH and greater LA concentration in treated silage”.

  1. Line-298-299: BC alone did not significantly reduced AA, see Table 2. Please rephrase.

Response: We have corrected: “After 14 d of ensiling, when LP and BC were used in combination decreased AA concentrations in treated silages”.

  1. Line-302-303: do you mean yeasts? Since molds were not found (Table 3)

Response: We are very sorry that this refers to yeasts. It has been corrected.
